# Antimicrobial and Cytotoxic Activities of the Secondary Metabolites of Endophytic Fungi Isolated from the Medicinal Plant *Hyssopus officinalis*

**DOI:** 10.3390/antibiotics12071201

**Published:** 2023-07-18

**Authors:** Farkhod Eshboev, Marina Karakozova, Jaloliddin Abdurakhmanov, Khayrulla Bobakulov, Khayotjon Dolimov, Akhror Abdurashidov, Asadali Baymirzaev, Artyom Makhnyov, Ekaterina Terenteva, Sobirdjan Sasmakov, Galina Piyakina, Dilfuza Egamberdieva, Pavel A. Nazarov, Shakhnoz Azimova

**Affiliations:** 1S.Yu. Yunusov Institute of the Chemistry of Plant Substances, Academy of Sciences of the Republic of Uzbekistan, 77 Mirzo Ulugbek Str., Tashkent 100170, Uzbekistan; ibrohimxoja@gmail.com (J.A.); khayrulla@rambler.ru (K.B.); hayotbekdolimov6272@gmail.com (K.D.); ahrorabdurashidov93@gmail.com (A.A.); asadali.baymirzaev@gmail.com (A.B.); artyom_alex@rambler.ru (A.M.); terenteva_katerina@bk.ru (E.T.); sasmakov@web.de (S.S.); gpiyakina@mail.ru (G.P.); genlab_icps@yahoo.com (S.A.); 2School of Chemical Engineering, New Uzbekistan University, 54 Mustaqillik Ave, Tashkent 100007, Uzbekistan; 3“Tashkent Institute of Irrigation and Agricultural Mechanization Engineers” National Research University, 39 Kori Niyoziy Str., Tashkent 100000, Uzbekistan; egamberdievad@gmail.com; 4Belozersky Institute of Physico-Chemical Biology, Lomonosov Moscow State University, 119991 Moscow, Russia; nazarovpa@gmail.com; 5Medical School, Central Asian University, Tashkent 111221, Uzbekistan

**Keywords:** antibiofilm activity, antimicrobial activity, *Chaetomium elatum*, cytotoxic activity, endophytic fungi, *Hyssopus officinalis*, MBC values, MIC values, secondary metabolites

## Abstract

According to the World Health Organization, it is estimated that by 2050, drug-resistant infections could cause up to 10 million deaths annually. Therefore, finding a new generation of antibiotics is crucial. Natural compounds from endophytic fungi are considered a potential source of new-generation antibiotics. The antimicrobial and cytotoxic effects of ethyl acetate extracts of nine endophytic fungal isolates obtained from *Hyssopus officinalis* were investigated for bioassay-guided isolation of the natural compounds. An extract of isolate VII showed the highest antimicrobial activities against Gram-positive bacteria *Bacillus subtilis* and *Staphylococcus aureus* (30.12 ± 0.20 mm and 35.21 ± 0.20 mm) and Gram-negative bacteria *Escherichia coli* and *Pseudomonas aeruginosa* (30.41 ± 0.23 mm and 25.12 ± 0.25 mm) among the tested extracts of isolates. Molecular identification of isolate VII confirmed it as *Chaetomium elatum* based on sequencing of its ITS genes, and it was discovered that this was the first time *C. elatum* had been isolated from *H. officinalis.* This isolate was cultured at a large scale for the isolation and identification of the active compound. Penicillic acid was isolated for the first time from *C. elatum* and its chemical structure was established by NMR spectroscopy. The penicillic acid showed strong antibacterial activities against *Bacillus subtilis* and *Staphylococcus aureus* with 20.68 mm and 25.51 mm inhibition zones, respectively. In addition, MIC and MBC values and antibiofilm activities of penicillic acid were determined. It was found that penicillic acid reduced the level of biofilms in proportion to antibacterial activity.

## 1. Introduction

Antibiotic resistance is an important issue due to the frequent use of antibiotics for the treatment of common bacterial infections. The enhancement of antimicrobial resistance is strengthening the pathogenicity and virulence of infectious microbes [1,2,3]. Therefore, finding new antibiotics is crucial to combat these resistant bacteria [4]. Natural compounds from endophytic fungi are potential candidates for new antibiotics against pathogenic bacteria [5]. Because secondary metabolites of endophytic fungi have been reported to exhibit a number of interesting potential biological activities such as antibacterial, antifungal, cytotoxic, antivirus, anti-inflammatory, etc. [6,7]. For example, three new polyketides, trichodermaketone E (1), 4-epi-7-O-methylkoninginin D (2), and trichopyranone A (3), and two new terpenoids, 3-hydroxyharziandione (4) and 10,11-dihydro-11-hydroxycyclonerodiol (5), were isolated from the endophytic fungus *Trichoderma koningiopsis* QA-3 obtained from *Artemisia argyi*. Compounds **1**, **3**–**5**, displayed inhibition against *E. coli*, with MIC values ranging from 0.5 to 64 µg/mL. Among the compounds tested, compound 3-hydroxyharziandione showed the strongest activity against *E. coli*, with a MIC value of 0.5 µg/mL [8]. In general, natural compound isolated from endophytic fungi are tested to antimicrobial and cytotoxic activities. There are several reasons for exploring the antimicrobial and cytotoxic activities of potential new drugs. Firstly, there is an urgent need for new drugs to treat both infectious diseases and cancer. Secondly, any new drug should be tested for both activities to ensure its safety and efficacy [9].

Recently, quite a lot of work has appeared on the isolation of penicillic acid from various sources, for example from Antarctic marine sediment or from various fungi of the genus of *Penicillium* [10,11]. Penicillic acid had antibacterial activity against Gram-positive and Gram-negative bacteria [12]. Although penicillic acid has been known since the 1930s [13], so far, its mechanism of action remains poorly understood. Although it is believed that penicillic acid may be an inhibitor of quorum sensing signals [14], the anti-biofilm effect has been shown only in a number of cases over a narrow concentration range [15].

Medicinal plants possess a distinct microbiome attributed to their unique bioactive secondary metabolites [16]. Therefore, in recent years, more attention has been given to the endophytic fungi of medicinal plants, which represent a rich source of new and useful natural compounds of interest to the pharmaceutical and agricultural industries and a potential source for the discovery of novel microorganisms [17]. Endophytic fungi inhabit plant tissues without instigating any noticeable symptoms of infection and do not cause adverse effects on their hosts [18,19,20]. To date, the endophytic fungi of only a few plants have ever been completely studied, like *Taxus brevifolia*. Taxol, an anti-cancer drug, was extracted from the endophytic fungus *Taxomyces andreanae* obtained from a plant. Thus, it is vital to study the diversity and chemical contents with biological activities of the endophytic fungi of medicinal plants [21,22].

*H. officinalis* is a polymorphous species that grows in central Asia, Iran, India, Mongolia, Europe, and Russia. This plant is used in folk medicine for the treatment of various diseases such as bacterial and fungal infections, wounds, asthma, cough, loss of appetite, and spasmodic disorders [23,24,25,26]. However, the diversity, chemical contents, and biological activities of the endophytic fungi of the *H. officinalis* have not been explored. Therefore, the aim of the research was the isolation of the endophytic fungi from the medicinal plant *H. officinalis*, the study of antimicrobial and cytotoxic activities of the fungal isolate’s extracts, and bioassay-guided isolation and identification of the active compound from the most active isolate. Moreover, studying the impact of the isolated active compound to bacterial cells and levels of biofilms formation properties of the bacteria.

## 2. Results

### 2.1. Isolation and Biological Activities of Ethyl Acetate Extracts of the Endophytic Fungi

Four Petri dishes with antibiotic-containing PDA (potato dextrose agar) medium were used for the isolation of endophytic fungi from stem and leaf samples of *H. officinalis*. As a result, a total of nine morphologically different endophytic fungi were isolated and obtained isolates were cultured in small volumes and extracted with acetate extracts (EtOAc). After that, the primary antimicrobial and cytotoxic activities of the extracts were determined in order to select the most active isolate among them for further work.

According to the results of antimicrobial activities, all isolates’ extracts except isolate V showed antimicrobial activities against the tested bacterial strains. The EtOAc extracts of isolate I, II, IV, VI, VIII and IX showed appreciable antibacterial activities against the tested bacterial strains. The extracts of the isolate III and VII manifested pronounced and strong antibacterial effects against the bacterial strains. Antibacterial activities of the extracts of isolate VII against Gram-positive bacteria *Bacillus subtilis* and *Staphylococcus aureus* were 30.12 ± 0.20 mm and 35.21 ± 0.20 mm, respectively, which were higher than the positive control ampicillin. The extracts of the three fungal isolates (II, VII, and VIII) showed activities against also *Candida albicans*. Isolate VII exhibited the highest antimicrobial activities among the obtained isolates (Table 1). Therefore, isolate VII was selected for further work for the isolation of antimicrobial active compounds and molecular identification of the isolate.

In vitro cytotoxic activities of the extracts of the isolates were tested against epithelial carcinoma of the cervix (HeLa), adenocarcinoma of the larynx (HEp-2), breast adenocarcinoma (HBL-100), T-lymphoblastic leukemia (CCRF-CEM), and normal hepatocyte and fibroblast cells. According to the results of cytotoxic activities, the extract of isolate III inhibited 98.5 ± 0.8% and 89.2 ± 0.5% of HEp-2 and HeLa, respectively. Additionally, the extract of isolate VIII showed 100.0 ± 0.0% and 77.5 ± 0.4% activities against CCRF-CEM and HeLa cell lines, respectively. However, both of the fungal extracts also inhibited normal cell lines, like 69.8 ± 0.4% and 100 ± 0.0% inhibition of fibroblast cells by extracts of isolate III and isolate VIII, respectively. Isolate VII also showed weak cytotoxic activity to HEp-2 and HeLa cancer cell lines and normal cells. Moreover, isolate IV for HBL-100 and isolate IX for HEp-2 and HeLa showed a proliferation effect (Table 2). Therefore, we have not cultured any isolates in large-scale cultivation for the isolation of cytotoxic active compounds.

### 2.2. Molecular Identification of Endophytic Fungus

Only isolate VII was molecularly identified among the obtained isolates since the antimicrobial activity of its extract showed the highest antimicrobial activity. At the same time, no proliferation of cancer cells or toxic effects on normal cells were shown. Therefore, we choose this isolate for further research such as molecular identification, isolation of the secondary metabolites, and their antimicrobial activities. Molecular identification of isolate VII was performed by sequencing ITS 4 and 5 regions of the fungus genome and the obtained DNA sequences (578 bp) were submitted to the GenBank with the accession number OP476344.1. Isolate VII was identified as a *C. elatum* according to a BLAST search in the NCBI database. In this study, *C. elatum* was isolated from *H. officinalis* for the first time. To date, endophytic fungi such as *Alternaria alternata* and *Cercospora* sp. were isolated from *H. officinalis* [27]. Additionally, *C. elatum* has been isolated from *Salvia officinalis* among the plants of the family of *Lamiaceae* [28]. The phylogenetic tree was constructed after the basic local alignment search tool (BLAST) search in the National Center for Biotechnology Information (NCBI) database using obtained ITS sequence together with the retrieved first nine sequences from the GenBank by MEGA 11 software (Figure 1) [29].

### 2.3. Bioassay-Guided Isolation and Establishment of the Structure of the Most Active Secondary Metabolite

In this research, we extracted the most active compound produced by *C. elatum*. The active compound was isolated from the acetone fraction using chromatographic techniques and isolated compound constituted 3% of the initial extract. The Rf value for compound **1** was 0.62.

The chemical structure of compound **1** was established based on the analysis of 1D (^1^H and ^13^C) and 2D (HSQC HMBC, and COSY) NMR spectra (Appendix A). The HSQC and COSY experiments were used to assign the signals of the proton and carbon nuclei in the ^1^H and ^13^C NMR spectra. In the highfield part of the ^1^H NMR spectrum of compound **1**, singlet signals of two methyl groups were found at δ_H_ 1.64 (H-7) and 3.82 (H-8). Further in the spectrum there are three broadened singlet signals at δ_H_ 5.18 (H-2), 5.07 (H-6*a*), and 5.31 (H-6*b*). According to the HSQC spectra, the last two signals belong to the exomethylene group.

Analysis of the ^13^C NMR and HSQC data of compound **1** showed the presence of eight resonance signals, representing 2 methyl, 1 methylene, 1 methine, and 4 quaternary. The downfield part of the ^13^C NMR spectrum contains two signals at δ_C_ 181.61 (C-3) and 173.38 (C-1). Further in the spectrum, signals are observed characteristic of the exomethylene group at δ_C_ 141.52 (C-5) and 116.38 (C-6), as well as signals of the quaternary carbon atom and the methine group at δ_C_ 104.74 (C-4) and 89.94 (C-2), respectively. In a more highfield part of the spectrum were observed signals of two methyl groups, one of which is connected with an oxygen atom at δ_C_ 60.55 (C-8) and 17.52 (C-7). The positions of the functional groups were established on the basis of the HMBC experiment data. In the spectrum of HMBC were observed cross-peaks between H-2/C-1, C-4; H-6/C-4, C-5, C-7; H-7/C-4, C-5, and C-6; H-8/C-3. This indicates that the ester group was located between C-2 and C-4, the exomethylene group at C-4, the methyl group at C-5, and at the C-3 position there was a methoxy group. The value of the chemical shift of the carbon signal at δ_C_ 104.74 (C-4) made it possible to establish the position of the hydroxyl group in this carbon. Detailed data of the ^1^H and ^13^C NMR spectra, as well as the HMBC correlation of compound **1**, are shown in Table 3. Thus, based on the above data and subsequent comparison with those of the literature data [30,31], the chemical structure of compound **1** was found to be penicillic acid (Figure 2).

### 2.4. Antimicrobial Activities of Penicillic Acid

The penicillic acid showed antibacterial effects against all tested bacteria (Figure 3). Penicillic acid exhibited strong antibacterial activities against Gram-positive bacteria, *Bacillus subtilis* and *Staphylococcus aureus* with 20.68 mm and 25.51 mm inhibition zones, respectively. Additionally, penicillic acid showed pronounced effects against Gram-negative bacteria, *Escherichia coli* and *Pseudomonas aeruginosa* with 20.45 mm and 15.61 mm inhibition zones, respectively. Penicillic acid exhibited a greater inhibition zone against *Escherichia coli* compared to the reference antibiotic gentamicin (18.61 mm). Ampicillin, gentamicin, and fluconazole were used as a positive control for Gram-positive bacteria, Gram-negative bacteria, and fungi, respectively (Figure 3). It should be noted that penicillic acid did not show an antimicrobial effect against *Candida albicans* (Figure 3), and even the extract of isolate VII which these compounds were isolated from showed strong antifungal activity with a 22.26 ± 0.15 mm inhibition zone (Table 1).

Moreover, to evaluate the minimum inhibitory concentration (MIC) and minimum bactericidal concentration (MBC) of the isolated penicillic acid, we used standard laboratory strains of *E. coli*, *M. smegmatis*, and *B. subtilis*. The MIC and MBC for *E. coli* and *B. subtilis* were almost the same, while for *M. smegmatis* the MIC differed by a factor of two, and the MBC was not reached.

Deletion mutants for TolC transporters (ΔAcrB, ΔAcrD, ΔAcrF, ΔMdtB, ΔMdtF, ΔMacB, ΔEmrB, and ΔEmrY) also show no significant difference in MICs. An insignificant difference was observed only in the deletion mutant for the TolC protein (Table 4).

### 2.5. Antibiofilm Activities of Penicillic Acid

Since penicillic acid had antibacterial activity against Gram-positive and Gram-negative bacteria, it would be interesting to know whether it could inhibit the growth of biofilms.

Already at 50 µM of penicillic acid, an almost threefold decrease in optical density (OD620) of the culture was observed, which then slowly decreases with increasing concentration of penicillic acid. Similar dynamics was demonstrated by biofilms that follow the change in optical density at 620 nm (Figure 4). This suggests that penicillic acid suppresses both biofilms and planktonic forms.

## 3. Discussion

Some biologically active compounds isolated from medicinal plants also isolated from endophytic fungi of these plants [32]. For instance, the anticancer compound taxol, which was originally isolated from the Pacific yew *Taxus brevifolia*, has also been found in the fungus *Taxomyces andreanae*, which was isolated from the same plant [33]. *H. officinalis* is also a medicinal plant widely used in folk medicine. According to Rusanova et al. [24], they isolated a total of 12 fungal isolates from *H. officinalis*, including eight *Alternaria alternata*, two *F. oxysporum, Bipolaris* sp., and *Epicoccum nigrum*. However, the biological activities of microorganisms isolated from this plant have not been studied. In this study, nine endophytic fungal isolates were obtained and their biological activities were investigated. Among these isolates, the most active one was molecularly identified as a *C. elatum*. Until now, *C. elatum* has not been isolated from *H. officinalis*. Some strains of *C. elatum* have been found to produce secondary metabolites with potential pharmaceutical applications, including compounds with anticancer, antiviral, and antifungal properties [28].

*C. elatum* was isolated from *H. officinalis* for the first time, and the biologically active compound penicillic acid was subsequently isolated and its structure was determined. Penicillic acid is a mycotoxin produced by several species of fungi, including *Penicillium species*, *Fusarium solani*, *Aspergillus species*, and others [31,34]. In this study, a new producer of penicillic acid has also been identified, which can synthesize it in large quantities. According to Zheng et al. [31], penicillic acid showed strong cytotoxic activities against six human cancer cell lines with IC_50_ values, 0.80–7.46 μg/mL, even though the compound was considered cancerogenic. Due to its low LD_50_ of 100 mg/kg (subcutaneous) for mice, penicillic acid has been found to be too toxic for therapeutic use. Therefore, in this study, more attention was given to the antibacterial properties of penicillic acid.

Penicillic acid has also shown antimicrobial properties and may contribute to the ability of certain fungi to compete with other microorganisms for resources [35]. A comparison of antibacterial activity against Gram-positive and Gram-negative bacteria indicates that penicillic acid seems to easily penetrate into the periplasmic space of Gram-negative bacteria, which is not surprising due to its small size and hydrophilicity. Thus, the main barrier to penicillic acid is the inner bacterial membrane. Since, apparently, the equilibrium of concentrations outside the cell envelope and in the periplasmic space is established rather quickly, we cannot observe the work of the MDR pumps to pump out penicillic acid outside the cell envelope. At the same time, this does not mean that the pumps are not involved in pumping out this substance, there is simply no difference in pumping it into the periplasmic space or outside the cell envelope.

Another important point may be the multiplicity of MDR pumps involved in pumping it out. This is indirectly confirmed by the fact that the deletion of the TolC protein, which makes 8 MDR pumps inactive at once [36], but slightly reduces the MIC. That is, multiple shutdowns of the pumps involved in pumping out can be detected, but it is not possible to determine which pumps are involved.

At the same time, it should be noted that the complex bacteria of the *Mycobacteriales* order are more resistant and penicillic acid does not have a bactericidal effect on them. This agrees with the data obtained earlier in [37] on the bacteria *Clavibacter michiganensis*. At the same time, the mechanism of resistance to penicillic acid is poorly understood, and apparently it may be more common in bacteria associated with plants.

Although penicillic acid is known as one of the inhibitors of quorum sensing signals [14,38], there is evidence that, in combination with EDTA and Patulin, it affects the formation of biofilms of dental unit water line isolates [39]; however, the observed effects were contradictory. In our experiments, we have shown that penicillic acid equally inhibits the growth of both planktonic and sessile parts of the *E. coli* bacterial population. We did not find any specific effect on the reduction of biofilm formation. However, the overall level of biofilms decreases in proportion to the decrease in the number of bacteria in the population due to the antibacterial action of penicillic acid.

In summary, penicillic acid can inhibit growth and be a bactericidal antibiotic against Gram-negative and Gram-positive bacteria, with the exception of *Mycobacteriales* order bacteria, as well as reduce the level of biofilms in proportion to antibacterial activity.

## 4. Materials and Methods

### 4.1. Materials

Components of the bacterial Luria–Bertani (LB) and Mueller–Hinton (MH) media were purchased from Helicon Company (Moscow, Russia) and HiMedia Laboratories (Mumbai, India). Ampicillin (10 μg/disk), ceftriaxone (30 μg/disk), gentamicin (10 μg/disk), and fluconazole (25 µg/disk) antibiotic contained disks were obtained from HiMedia Laboratories, (India). DMEM and RPMI-1640, FBS and L-glutamine for cell cultivation were purchased from Sigma-Aldrich (St. Louis, MO, USA). A BigDye Terminator v3.1 Cycle Sequencing Kit was purchased from Thermo Fisher Scientific (Waltham, MA, USA). Other reagents were from Sigma-Aldrich (St. Louis, MO, USA).

### 4.2. Bacterial Strains

*Bacillus subtilis* RKMUz—5, *Staphylococcus aureus* ATCC 25923, *Escherichia coli* RKMUz—221, *Pseudomonas aeruginosa* ATCC 27879, and *Candida albicans* RKMUz—247 were used for antimicrobial activities by disk-diffusion method. The RKMUz microorganism cultures were obtained from the strain collection of the Institute of Microbiology, Academy of Sciences of Uzbekistan.

Standard laboratory strains Escherichia coli Castellani and Chalmers 1919, strain MG1655 (F-lambda-ilvG-rfb-50 rph-1), and Bacillus subtilis subs. subtilis Cohn 1872, strain BR151 (trpC2 lys-3 metB10) were used. Mycobacterium smegmatis (entry #377) were obtained from the Microorganisms Collection of the Moscow State University.

Deletion strains ECK3026 (devoid of the tolC gene), ECK0456 (devoid of the acrB gene), ECK2465 (devoid of the acrD gene), ECK3253 (devoid of the acrF gene), ECK2071 (devoid of the mdtB gene), ECK3498 (devoid of the mdtF gene), ECK0870 (devoid of the macB gene), ECK2680 (devoid of the emrB gene), and ECK2363 (devoid of the emrY gene) were kindly provided by Dr. H. Niki, National Institute of Genetics, Japan65.

The microorganisms used in this study, including *B. subtilis*, *M. smegmatis* and *C. abligans* were grown at 30 °C, while *E. coli*, *P. aueruginosa*, *S. aureus* were grown at 37 °C, all in LB medium with shaking at a frequency of 140 rpm [40].

### 4.3. Isolation of Endophytic Fungi

Stems and leaves of *H. officinalis* were collected from the Medicinal Plant section of the Botanical Garden at the Academy of Sciences of Uzbekistan. Initially, the plant materials were washed using sterile water to remove any external substances. Subsequently, the samples underwent surface sterilization through a series of steps. They were immersed in a 5% sodium hypochlorite solution for 1 min, followed by a 70% ethanol solution for 1 min, and finally rinsed with sterile distilled water. The stems and leaves were then cut into pieces measuring 0.5 cm × 0.5 cm after surface-drying and placed on potato dextrose agar (PDA) media supplemented with the antibiotic ampicillin (100 mg/L). The cultures were incubated at a temperature of 28 ± 2 °C until fungal growth became visible. Additionally, 0.2 mL samples from the final wash process of each plant material were placed on PDA medium and incubated at 28 °C for a week to confirm surface sterilization. Then, morphologically distinct fungal colonies were cultured separately on PDA medium and incubated for 7 days. After 7 days of incubation, the isolates were cultured several times on PDA to obtain final pure cultures.

### 4.4. Preparation of Endophytic Fungal Extracts

To culture the fungal isolates, active growing colonies of the isolates were transferred into a 500 mL Erlenmeyer flask containing 200 mL of potato dextrose broth (PDB). The flask was then placed on a shaker at 28 °C and shaken continuously at 150 rpm for 14 days. Cultures of fungal isolates were extracted five times with an equal volume of EtOAc. The EtOAc extracts were evaporated under reduced pressure at 40 °C, using rotary evaporator to obtain crude EtOAc extracts.

### 4.5. Antimicrobial Activities of Extracts

The antimicrobial activity of the extracts obtained from endophytic fungal isolates derived from *H. officinalis* was evaluated using the agar disk-diffusion method, as described in reference [41]. The antimicrobial activity was assessed against five species of microorganisms, including Gram-positive bacteria *B. subtilis* and *S. aureus*, Gram-negative bacteria *E. coli* and *P. aeruginosa*, and the yeast *C. albicans*. To create a solid medium, sterile nutrient agar was prepared by dissolving 28 g of agar in one liter of distilled water. Bacterial cells were inoculated into the medium by adding 200 µL of bacterial cells suspended in 2 mL of 0.9% NaCl solution to 25 mL of the nutrient agar. The mixture was thoroughly mixed and then poured into petri dishes, allowing it to solidify. *C. albicans*, with a concentration of 1 × 10^6^ colony forming units (CFU) per mL, was inoculated into sterile Mueller–Hinton agar. Paper disks with a diameter of 6 mm, made of sterile Whatman No.1 filter paper, were used for applying the EtOAc extracts of fungi at a concentration of 2 mg per disk (or 200 µg per disk for isolated compounds). Positive controls, including ampicillin (10 μg/disk), ceftriaxone (30 μg/disk), gentamicin (10 μg/disk), and fluconazole (25 µg/disk), were also applied on separate disks. The negative controls consisted of disks treated with solvents only. After applying the substances onto the paper disks, the solvents were allowed to evaporate by placing the disks in a stream of air. Subsequently, the disks were carefully placed on the surface of agar plates that had been previously inoculated with the respective microorganisms. Plates containing bacteria were then incubated at 37 °C for 24 h, while plates with *C. albicans* were incubated at 28 °C for 48 h. Following the incubation period, the diameter of the inhibition zone (including the disk diameter) was measured and recorded. This process was repeated three times in independent assays, and an average zone of inhibition was calculated from the three replicates.

### 4.6. Cytotoxic Activities

Cervical adenocarcinoma HeLa, larynx carcinoma HEp-2 (ATCC:CCL-23) and breast cancer HBL-100 (ATCC HTB 124) cells were purchased from the Bank of Cell cultures in the Institute of Cytology of Russian Federation, T-lymphoblastic leukemia CCRF-CEM (ATCC: CCL-19) cells were obtained from the University of Heidelberg, Germany. Cell lines were cultured in advanced DMEM and RPMI-1640 supplemented with 10% inactivated FBS and 2 mM L-glutamine and 1% antibiotic-antimycotic solution, and grown at 37 °C in a humidified atmosphere of 5% CO_2_ (SHELLAB, Cornelius, NC, USA) [42].

Fibroblast cells were obtained by the method in [43]. For this, skin biopsies of a newborn rat were crushed, washed with a 1% solution of an antibiotic-antimycotic (Himedia, India), and kept in a 5% solution of trypsin-EDTA, after which the cells were passed through a Teflon filter. Then, the cells were centrifuged at 800 rpm for 8 min. The pellet of fibroblasts was placed in culture flasks with a DMEM/F-12 medium with 20% FBS and incubated at 37 °C, 5% CO_2_. After 48 h, the culture medium was changed. After 72 h, the cells were ready for further use.

To obtain hepatocyte cells, a rat liver was extracted in vivo, crushed and homogenized, being careful not to destroy the cell walls. Next, a 0.9% NaCl solution was added to the homogenate and centrifuged at 800 rpm. The pellet was resuspended in DMEM growth medium using 10% FBS, 1% antimycotic antibiotic solution, and 2 mM glutamine solution. Cells were cultured in 96-well plates in a CO_2_ incubator at 37 °C, 5% CO_2_, and 80% medium. After 72–100 h, the cells were ready for further use [44].

All extracts were dissolved in DMSO (0.8% by volume) immediately before the experiment. The cytotoxic properties of the secondary metabolites of the fungi were determined by the MTT method in vitro. For that, cancer cells were seeded at a concentration of 2 × 10^4^ cells per 1 mL of medium in 96-well plates, 100 μL per well; healthy cells were seeded at a concentration of 5 × 10^4^ cells per 1 mL of medium and incubated for 24 h in a CO_2_ incubator. Next, the nutrient medium was removed, and fresh medium with test samples at a concentration of 100 μg/mL was added to the wells to the cells. For the initial study of cytotoxicity, we chose the maximum possible concentration of test samples of 100 μg/mL, since higher concentrations give nonspecific cytotoxicity. Incubation of cells with extracts continued for 24 h. Next, 20 µL of MTT solution (5 mg/mL) (Acrosorganics, Belgium) was added to the cells and incubated for 3–4 h. After that, the wells were emptied, and 50 µL of DMSO was added. The optical density was determined at 630 nm in a plate reader. Cell viability was determined by the ratio of living cells exposed to the test samples to the number of living cells in the control. The well-known anticancer drug Cisplatin-Naprod (India) in 100 μM concentration was used as the reference drug.

### 4.7. Molecular Identification

The internal transcribed spacer (ITS) region of the ribosomal gene was sequenced for molecular identification. Fungal DNA was isolated by a modified hot phenol method using guanidine [45]. After isolation, the obtained DNA was used in a volume of 10 μL for polymerase chain reaction (PCR) analysis. Then, PCR analysis was conducted for amplification of ITS region using the following primers: the forward primer ITS4 (5′ TCCTCCGCTTATTGATATGC3′); the reverse primer ITS5 (5′GGAAGTAAAAGTCGTAACAAGG 3′) [5]. The PCR was performed in 25 μL reaction volume comprising 5 μL of 5× reaction buffer (25 мM (NH_4_)_2_SO_4_, 187.5 мM KCl, 150 mM tris buffer pH 8,5), 1.5 μL of 50 mM MgCl_2_, 0.5 μL of 10 mM dNTP’s, 0.3 μL of 20 μmol primers, 0.5 μL M-Mlv 50 U/μL, 0.5 μL SynTaq 5 U/μL, 6.4 μL of distilled water, and 10 μL of DNA. The temperature profile was maintained as follows: initial denaturation at 95 °C for 5 min, followed by 35 cycles of 95 °C for 30 s, 60 °C for 30 s, and 72 °C for 1 min, and final elongation at 72 °C for 5 min. The results of PCR were analyzed by gel electrophoresis in 3% agarose gel. After carefully staining the gel with ethidium bromide, the amplified DNA fragments were observed under UV light transilluminator. The ITS region of the ribosomal gene was sequenced using a BigDye Terminator v3.1 Cycle Sequencing Kit (Thermo Fisher Scientific, Waltham, MA, USA) and universal primers (ITS4 and ITS5) [46]. Seven identical cycle sequencing reactions were conducted in 10 µL containing 20 ng of the cDNA, 10 pmol of the gene-specific primers, 4 µL of BigDye Terminator v3.1 ready reaction mix, and deionized water to a final vol of 10 µL. Temperature mode was installed as follows: initial denaturation step at 96 °C for 2 min, followed by 30 cycles of 96 °C for 20 s, 52 °C for 20 s, and 65 °C for 4 min in a QuantStudio PCR System (Applied Biosystems, Foster City, CA, USA, Thermo Fisher Scientific, Waltham, MA, USA). Sequencing purification was performed using the n-butanol method [39]. The sequencing was performed in a SeqStudio Genetic Analyzer (Applied Biosystems, Thermo Fisher Scientific, Waltham, MA, USA) in our laboratory. The obtained sequences were analyzed using the National Center for Biotechnology Information (NCBI) GenBank database and submitted to NCBI. Phylogenetic analysis was carried out by using the neighbor-joining method with bootstrap support of 1000 replicates using MEGA 11 software.

### 4.8. Large Scale Cultivation and the Bioassay-Guided Secondary Metabolite Isolation

After the brief biological activities of extracts of isolated endophytic fungi, only one isolate VII was cultivated in a large scale. To scale up the cultivation of isolate VII, an actively growing colony was first cultivated in a single flask, then transferred to 20 separate 500 mL Erlenmeyer flasks containing potato dextrose broth (PDB). The flasks were then incubated at 28 °C with continuous shaking at 150 rpm for 14 days to allow for the growth and production of the desired compound [47]. After cultivation, the fungal culture was extracted with EtOAc and evaporated using the rotary evaporator. The obtained extract was washed in an isocratic system with hexane, chloroform, EtOAc, acetone, and methanol in order to increase the polarity of the solvents in a silica gel column. After that, each fraction was tested against conditionally pathogenic microorganisms and the acetonic fraction was the most active. Therefore, we continued working on this fraction for the isolation of the most active compound. The acetonic fraction was separated using double chromatographed by the Sephadex LH 20 in methanol and the bioactive compound of interest was isolated. The level of purity of the fractions was tested by the thin layer chromatography (TLC) method in different solvent systems.

### 4.9. NMR Spectroscopy Analysis

NMR spectra were recorded on a JNM-ECZ600R spectrometer (JEOL, Tokyo, Japan, 600 MHz for ^1^H and 150 MHz for ^13^C) in CD_3_OD. TMS (*δ* 0.00 ppm) was used as an internal standard for ^1^H NMR shifts, and solvent signal (CD_3_OD, 49.00 ppm vs. TMS) was used as a reference for ^13^C NMR shifts. NMR spectra were processed using MestReNova 14.2.0 software (Mestrelab Research S.L., Santiago de Compostela, Spain).

### 4.10. Bacterial Growth Suppression Assay

Overnight bacterial cultures were diluted in fresh LB media. A 200 μL aliquot of bacterial cell cultures (5 × 10^5^ cells/mL) was inoculated into 96-well plates (Eppendorf AG, Hamburg, Germany) and penicillic acid was added at concentrations of 50, 100, 200, 400, 800, 1600, and 2000 μM. Every concentration was tested in 4 wells parallelly. The cells were allowed to grow for 21 h at 37 °C. Optical densities at 620 nm were measured with a Thermo Scientific Multiskan FC (Thermo Fisher Scientific, USA) equipped with incubator. All experiments were carried out at least in triplicate.

### 4.11. MIC and Minimal Bactericidal Concentration (MBC) Determination

MICs for penicillic acid were determined, as recommended by CLSI [48], by Mueller–Hinton broth microdilution using in-house-prepared panels. The compounds were diluted in a 96-well plate to final concentrations ranging from 5 to 2000 μM in 250 mL aliquot of the bacterial suspension (5 × 10^5^ CFU/mL) followed by the incubation for 18 h at 37 °C. MIC was defined as the lowest concentration completely inhibiting bacterial growth, which was visually detected along with CFU and OD measurements. Experiments were carried out in triplicate.

MBC was determined as the lowest concentration at which no colonies were grown during CFU measurement.

### 4.12. Biofilm Development

To assess how compounds affect biofilm formation, we used the protocol [49] and evaluated the effect of a substance on biofilm formation as a change in the ratio between planktonic and sessile forms of bacteria.

The bacterial biofilms were cultivated in LB media in polystyrene 96-well plates (Citotest, Nantong, China). A panel of different concentrations of penicillic acid was prepared by the double dilutions method, an aliquot of 200 µL was added to each well. No antibiotics were added to control samples. *E. coli* cell suspension (5 × 10^5^ cells per mL) was added to each well. Microtiter plates were incubated for 20 h at 37 °C in a Thermo Scientific Multiskan FC plate reader with an incubator (Thermo Fisher Scientific, Waltham, MA, USA). Bacterial growth was observed by means of OD620 measurements. After incubation, the liquid medium was removed by turning the plate upside down to drain the cell suspension, and the plate was washed three times with PBS buffer to remove free cells. To fix the remaining biofilms, the plates were placed in a thermostat at 60 °C for 30 min. To stain biofilms, 40 µL of 1% crystal violet was added to each well and left to stain for 30–60 min. The crystal violet was discarded, the plate was washed twice with deionized water, then 200 mL of 95% ethanol was added to each well to extract the crystal violet for 3 h. Measurement of the absorbance of crystal violet was performed by means of OD595 using the microplate reader. All biofilm samples were prepared in quadruplicate.

### 4.13. Statistical Analysis

Origin 8.6 software (Microcal Software Inc., located in Northampton, MA, USA) was used to conduct statistical analysis and exponential curve fitting. The outcomes were presented as the mean value plus the standard error (SE). A one-way ANOVA test was conducted to assess the statistical significance of the findings.

## 5. Conclusions

Endophytic fungus *C. elatum* was isolated for the first time from *H. officinalis*. EtOAc extract of the *C. elatum* showed the most effective antimicrobial activities among the extracts of the obtained fungal isolates. It was found that *C. elatum* extract showed strong antibacterial activity due to mycotoxin penicillic acid. Penicillic acid was first isolated from the fungus *C. elatum*, and its structure was established by NMR spectroscopy. This result provides further evidence for the synthesis of specific substances by different organisms. However, it raises the question of whether these substances are synthesized uniformly across all organisms. To address this question, future studies should focus on investigating the biochemical synthesis pathway of penicillic acid and identifying the gene clusters involved in its production. Penicillic acid showed strong antibacterial and antibiofilm activities. In addition, the high content of penicillic acid in *C. elatum* suggests that it could be a promising source of bioactive compounds for potential future applications, including the development of new, non-toxic active compounds through chemical modification. Therefore, our future work will focus on several tasks, such as investigating the mechanism of action of penicillic acid on bacterial cells, synthesizing its chemical derivatives, and evaluating its biological activity.

## Figures and Tables

**Figure 1 antibiotics-12-01201-f001:**
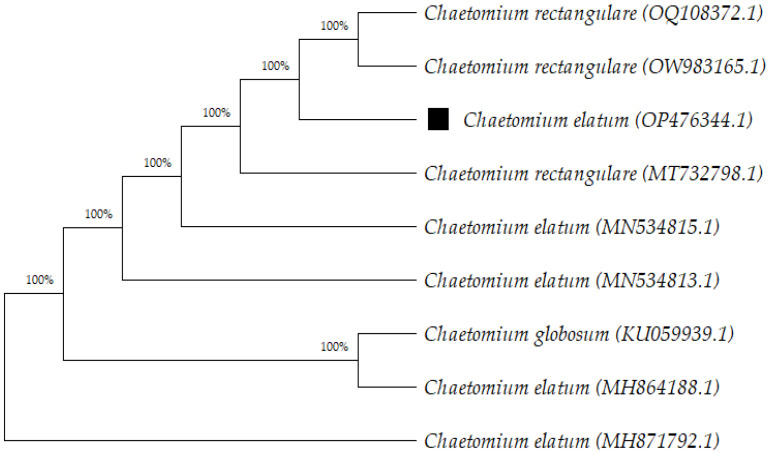
Neighbor-joining phylogenetic tree constructed from ITS (internal transcribed spacer) sequence of the isolate VII and ITS sequences obtained from the GenBank database after BLAST search by using MEGA 11 (Molecular Evolutionary Genetics Analysis, version 11). The bootstrap consensus tree was inferred from 1000 replicates. All positions containing gaps and missing data were eliminated from the dataset (complete deletion option).

**Figure 2 antibiotics-12-01201-f002:**
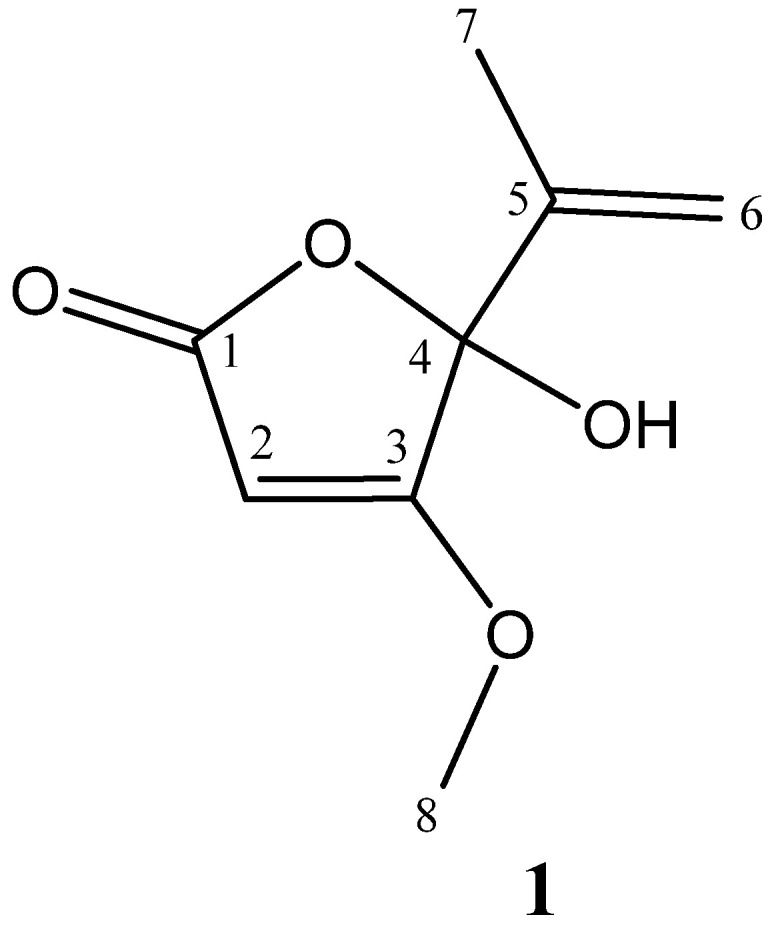
The chemical structure of the compound **1** (penicillic acid).

**Figure 3 antibiotics-12-01201-f003:**
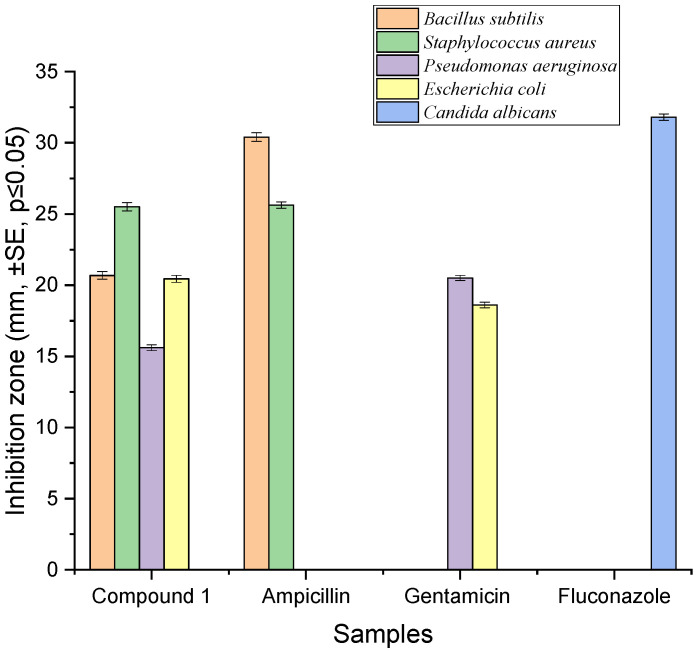
Antimicrobial activities of penicillic acid derived from *C. elatum*.

**Figure 4 antibiotics-12-01201-f004:**
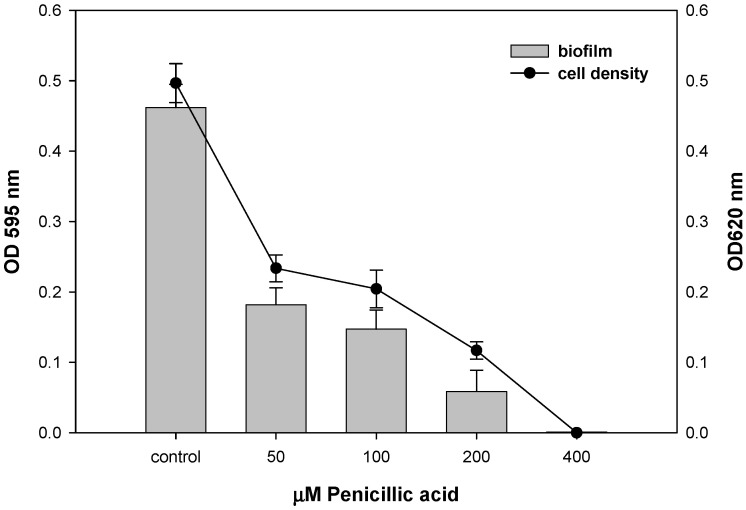
Biofilm formation of *E. coli* at different concentrations of penicillic acid (penicillic acid showed a similar impact on the biofilm formation properties of *B. subtilis*, so it is not given in the text but it is given in the Supplementary Material Appendix A). *E. coli* WT cells were exposed to 20 h with penicillic acid, and their cell density (black curves) and surface attachment (gray bars) were measured. Values for cell density (OD_620_) are indicated on the right *y*-axis, and biofilm values (OD_595_) are indicated on the left *y*-axis.

**Table 1 antibiotics-12-01201-t001:** Antimicrobial activities of EtOAc extracts obtained from fungal isolates of *H. officinalis*.

Fungal Isolates	Inhibition Zone (mm, ±SE, *p* ≤ 0.05)				
	Gram-Positive Bacteria		Gram-Negative Bacteria		Fungus
	*Bacillus subtilis*	*Staphylococcus aureus*	*Escherichia coli*	*Pseudomonas aeruginosa*	*Candida albicans*
Isolates I	12.03 ± 0.27	12.12 ± 0.27	10.26 ± 0.14	8.02 ± 0.28	NA
Isolates II	13.03 ± 0.28	14.22 ± 0.15	14.29 ± 0.17	12.25 ± 0.14	10.46 ± 0.26
Isolates III	21.12 ± 0.30	16.25 ± 0.14	14.42 ± 0.23	10.10 ± 0.26	NA
Isolates IV	8.11 ± 0.22	11.46 ± 0.26	6.22 ± 0.12	6.05 ± 0.26	NA
Isolates V	NA	NA	NA	NA	NA
Isolates VI	12.00 ± 0.27	10.62 ± 0.17	10.35 ± 0.00	7.25 ± 0.14	NA
Isolates VII	30.12 ± 0.20	35.21 ± 0.20	30.41 ± 0.23	25.12 ± 0.25	22.26 ± 0.15
Isolates VIII	12.37 ± 0.21	10.71 ± 0.27	10.60 ± 0.16	8.42 ± 0.25	10.31 ± 0.18
Isolates IX	11.18 ± 0.42	10.09 ± 0.43	7.10 ± 0.54	7.01 ± 0.53	NA
Ampicillin	27.42 ± 0.18	28.26 ± 0.21	NT	NT	NT
Ceftriaxon	NT	NT	30.14 ± 0.19	30.51 ± 0.13	NT
Fluconazol	NT	NT	NT	NT	31.24 ± 0.20

NA no activity; NT not tested.

**Table 2 antibiotics-12-01201-t002:** Cytotoxic activities of EtOAc extracts obtained from fungal isolates of *H. officinalis*.

Extracts	Inhibition of Cell Growth (%, ±SE, *p* ≤ 0.05)					
	CCRF-CEM	HEp-2	HeLa	HBL-100	Hepatocyte	Fibroblast
Isolate I	0.0 ± 0.0	4.5 ± 0.5	6.7 ± 0.5	0.0 ± 0.0	0.0 ± 0.0	0.0 ± 0.0
Isolate II	0.0 ± 0.0	47.0 ± 0.7	66.3 ± 0.6	0.0 ± 0.0	21.0 ± 0.0	26.1 ± 0.0
Isolate III	53.8 ± 0.8	98.5 ± 0.8	89.2 ± 0.5	64.3 ± 0.8	31.2 ± 0.5	69.8 ± 0.4
Isolate IV	8.3 ± 0.5	0.0 ± 0.0	14.0 ± 0.8	Proliferation	0.0 ± 0.0	0.0 ± 0.0
Isolate V	0.0 ± 0.0	29.8 ± 0.3	12.1 ± 0.3	0.0 ± 0.0	28.5 ± 0.6	19.6 ± 0.5
Isolate VI	22.4 ± 0.2	24.2 ± 0.5	24.0 ± 0.2	19.3 ± 0.1	18.8 ± 0.4	10.0 ± 0.0
Isolate VII	0.0 ± 0.0	32.4 ± 0.3	40.2 ± 0.2	0.0 ± 0.0	17.8 ± 0.2	19.03 ± 0.1
Isolate VIII	100.0 ± 0.0	0.0 ± 0.0	77.5 ± 0.4	21.2 ± 0.5	46.3 ± 0.5	100 ± 0.0
Isolate IX	0.0 ± 0.0	Proliferation	Proliferation	0.0 ± 0.0	0.0 ± 0.0	0.0 ± 0.0
Cisplatin	59.5 ± 0.5	84.4 ± 0.5	90.8 ± 0.4	100.0 ± 0.0	51.1 ± 0.5	44.0 ± 0.5

**Table 3 antibiotics-12-01201-t003:** ^1^H and ^13^C NMR and HMBC Data of compound **1** (*δ*, ppm, 600 MHz for ^1^H and 150 MHz for ^13^C) in CD_3_OD.

Atom C	δ_C_	δ_H_	HMBC (H→C)
1	173.38		
2	89.94	5.18, s	1, 4
3	181.61		
4	104.74		
5	141.52		
6	116.38	5.07, br.s	
		5.31, br.s	4, 5, 7
7	17.52	1.64, s	4, 5, 6
8	60.55	3.82, s	3

**Table 4 antibiotics-12-01201-t004:** Susceptibility of bacterial strains to penicillic acid: minimal inhibitory concentration (MIC) and minimal bactericidal concentration (MBC) measurements.

Bacteria	MIC		MBC	
	μM	µg/mL^−1^	μM	µg/mL^−1^
*Bacillus subtilis*	200	34	800	128
*Mycobacterium smegmatis*	400	68	>2000	>340
*Escherichia coli* ^1^	300	51	800 ^2^	128 ^2^
*Escherichia coli* ΔTolC	200	34	ND	

^1^ Including deletion strains ΔAcrB, ΔAcrD, ΔAcrF, ΔMdtB, ΔMdtF, ΔMacB, ΔEmrB, and ΔEmrY. ^2^ MBC for deletion mutants was not determined.

## Data Availability

Not applicable.

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
