# Peer review of "Antimicrobial and Cytotoxic Activities of the Secondary Metabolites of Endophytic Fungi Isolated from the Medicinal Plant Hyssopus officinalis"

_antibiotics, 2023, doi:10.3390/antibiotics12071201_

Round 1
Reviewer 1 Report
Authors investigated the antimicrobial and cytotoxic activity of penicillic acid extracted by an endophytic fungus isolated from the medicinal plant H. officinalis.
After an accurate reading of the paper I suggest to add an experiment to confirm the activity of penicillic acid against the tested bacteria. In detail, I suggest to buy synthetic penicillic acid and carry out the same experiments performed in this study.
The aim of the study is not fully clear to me. What it your purpose once the antimicrobial and cytotoxic activity of penicillic acid have been tested? Please also explain this in the introduction.
Since the paper is focused on penicillic acid extracted from C. elatum I would modify the title of the study with: "Antimicrobial and cytotoxic activities of penicillic acid of Chaetomium elatum isolated from the medicinal plant Hyssopus officinalis"
Please find below my comments, you also find a file as attachment:
INTRODUCTION: it should be improved, deeply exploring the role of antimicrobial and cytotoxic activity of endophytic fungi isolated from plants. Moreover, seen paragraph 2.1 and 4.4, what is the current use of fungal compounds in medicine should be addressed. Indeed. it sounds difficult to me to understand the reason why cytotoxic activity test was performed on human cells. Do you plan to test fungal extracts/fungal compounds in human medicine in the next future? Are these compounds toxic also for healthy cells? Please provide information in the Introduction section.
- line 39-42: please try to join the sentences.
RESULTS:
- please substitute Ch. elatum with C. elatum in the whole text.
- please substitute Gramm with Gram in the whole text and choose beween GRAM or Gram form.
- line 117: add bibliography for MEGA 11
- line 119: please add an outgroup to build the tree.
- line 129-132: sentence already present in mat&met
- line 172: what concentration for antibiotics?
- line 174: please write the name of the fungus.
- line 198: I suppose 50uM is referred to penicillic acid. If so, please clarify. Optical density of what? Please clarify.
- line 209-212: the highlighted sentence is not very clear and some errors are present. Please check.
- line 225-226: please join the 2 sentences in only one.
MATERIAL AND METHOD:
- please add the origin of the tested bacteria which is not specified
- line 276: how did you inoculate the fungus? A plug from an active growing colony or spore suspension? Please explain in the text. The same for line 346.
- line 289-291: it is not clear to me how the plates were prepared? Did you mixed bacterial cells in warm sterile nutrient agar? Please explain.
- line 294: what concentration of antibiotics? What company did you purchased themP Please explain in the text.
- line 313: what concentration of cisplatin? Please explain in the text
- line 351: please specify the names of pathogenic microrganisms
- line 365: maybe the word "stain" should be "strain"?
- line 374: Please specify what bacteria were grown at 30°C and what bacteria at 37°C. What time of incubation?
- line 379: please clarify how you have added penicillic acid: how many concentrations in how many wells?

I would suggest the authors to use an English service editing to improve the quality of the language.
Author Response
Dear Reviewer,
Thank you very much for your review! Your comments are very helpful for us in compiling and considering the manuscript, as well as for further studies.
Best wishes,

Reviewer 2 Report
The paper entitled Antimicrobial and cytotoxic activities of the secondary metabolites of endophytic fungi isolated from the medicinal plant Hyssopus officinalis is written well and is good idea to find natural compounds to fight antibiotic resistance, However I will suggest following changes.
1. Introduction needs modification. Elaborate the same.
2. Table 1 and 2 should be supported with additional data/images.
3. Analysis of the 13C NMR and HSQC data of compound 1 data missing. Add the data in supplementary.
4. Also, data about the chemical structure of compound 1 based on the analysis of 1D 134 (1H and 13C) and 2D (HSQC HMBC, and COSY) NMR spectra is missing. Add in supplementary.
5. No data about morphological differences on the basis of which the author isolated the endophytes.
6. Kindly see fig. 3 and correct it. Why was compound 1 not tested against candida?
7. In fig 4 why Biofilm formation was only checked for E. coli?
8. Add more to discussion.
9. The cytotoxic activities of the extracts of endophytic fungi were carried out against four cancer cell line. But how were the cell lines obtained/collected. How were they grown. What were the conditions. When and how was treatment given. Was compound tested against the cell lines. How wereconcentrations selected?
10. Conclusion should not be summary of work. Modify the same.
quality of english is acceptable
Author Response

(The authors gave the same response as above.)

Reviewer 3 Report
Article
Antimicrobial and cytotoxic activities of the secondary metabolites of endophytic fungi isolated from the medicinal plant Hyssopus officinalis
Abstract
… (30.12±0.20 mm and 35.21±0.20 mm) and Gram-negative (30.41±0.23 mm and
25.12±0.25 mm) bacteria
Key words: Antibiofilm activity; Antimicrobial activity; Chaetomium elatum; Cytotoxic activity; Endophytic fungi; Hyssopus officinalis; MBC values; MIC values; Secondary metabolites.
1. Introduction
2. Results
Page 2, line 66: … with antibiotic-containing PDA (Potato dextrose agar)
Line 78: … were 30.12±0.20 mm and 35.21±0.20 mm, respectively ….
Line 79: … than the positive control ampicillin.
Page 3, lines 91, 92, 93, 94, 95: … inhibited 98.5±0.8 % and 89.2±0.5 % of HEp-2 and HeLa, respectively; … showed 100.0±0.0 % and 77.5±0.4 % activities …; … and HeLa cell lines, accordingly. … like 69.8±0.4 % and 100±0.0 % inhibition of fibroblast cells by extracts of isolate III and isolate VIII, respectively.
Line 116: … after the BLAST (Basic Local Alignment Search Tool) search in the NCBI (National Center for Biotechnology Information) database …
Figure 1. … from ITS (Internal transcribed spacer) sequence …; … MEGA (Molecular Evolutionary Genetics Analysis, version 11)
Table 3.1. Improve this table.
Page 5, lines 168, 177: … inhibition zones, respectively. … 22.26±0.15 mm inhibition zone.
3. Discussion
1. Materials and methods
… ascorbic acid (AA) (> 99 % pure)…, and so on …
Line 143: … (mg GAE ·(100 g d.m.) -1)
Line 144: … (mg Trolox equivalent (TE)(100 g d.m.) -1)
Line 145: … (µmol Trolox equivalent (TE) (100 g d.m.) -1).
Table 4: µg/mL or μgmL-1
4. Materials and Methods
Page 8, line 265, 270: … in 5 % sodium hypochlorite solution for 1 min and in 70 % ethanol for …; … 0.2 mL samples …
Page 9, line 289: … (28 g agar/L distilled water) was inoculated with bacterial cells (200 µl of bacterial cells in 2 mL 0.9 % NaCl suspension and 25 mL …
Line 292: … (1×106 colony forming units (CFU) per mL) was …
Lines 299, 300: … for 24 h at 37 °C and ...
Line 306, 307: … in DMSO (0.8 % by volume) and …, … of 100 µg/mL …
Lines 308, 309: Next, 20 µL of MTT solution (5 mg/mL (Acrosorganics, Belgium) was added to the cells and incubated for 3–4 h. After that, the wells were emptied, and 50 µL of …
Line 317: … in a volume of 10 µL
Line 322: … (25 mМ (NH4)2SO4, 187,5 mМ KCl, 150 М tris buffer рН 8,5)
Page 10, line 336: … at 96 °C for 2 min, followed by 30 cycles of 96 °C for 20 s, 52 °C for 20 s, and 65 °C …
Page 11: (5 × 105 cells/mL)
Line 387: … 250 mL aliquot of the bacterial suspension (5 × 105 CFU/mL)
Line 406: … at 60 °C for 30 min. To stain biofilms, 40 μL of 1 % …
Author Response
Dear Reviewer,
Thank you very much for your review! Your comments are very helpful for us in compiling and considering the manuscript, as well as for further studies.
Best regards,

Round 2
Reviewer 1 Report
The manuscript is improved in every section and I thank authors for their replies and effort.
I have found some little errors in the text:
- line 64: penicillin acid should be penicillc acid;
- line 238: please check the sentence which is not clear
Good luck for your future investigations on this work.
In my opinion the English language quality is sufficient even if minor editing should be addressed
Author Response
Dear Reviewer,
Thank you very much for your comments on our manuscript!
They helped to improve qualities of our manuscript.
Best regards,
